# Effects of Silver Nanoparticles on Proliferation and Apoptosis in Granulosa Cells of Chicken Preovulatory Follicles: An In Vitro Study

**DOI:** 10.3390/ani11061652

**Published:** 2021-06-02

**Authors:** Dorota Katarzyńska-Banasik, Anna Kozubek, Małgorzata Grzesiak, Andrzej Sechman

**Affiliations:** 1Department of Animal Physiology and Endocrinology, University of Agriculture in Krakow, Al. Mickiewicza 24/28, 30-059 Krakow, Poland; anna.kozubek@urk.edu.pl (A.K.); andrzej.sechman@urk.edu.pl (A.S.); 2Department of Endocrinology, Institute of Zoology and Biomedical Research, Jagiellonian University in Krakow, Gronostajowa 9, 30-387 Krakow, Poland; m.e.grzesiak@uj.edu.pl

**Keywords:** silver nanoparticles, proliferation, apoptosis, caspase-3, chicken

## Abstract

**Simple Summary:**

The goal of this research was to examine the effect of silver nanoparticles (AgNPs) on the proliferation and apoptosis process in the in vitro cultured granulosa cells of chicken preovulatory follicles. The study revealed that both 13 nm and 50 nm AgNPs significantly repressed proliferation of the granulosa cells, and this effect was faster for 50 nm nanoparticles. Caspase-3 activity increased in granulosa cells under the influence of 13 nm AgNPs, while in the case of nanoparticles 50 nm in size, this result was not observed, which may suggest that AgNPs evoke different cell death pathways. In conclusion, these data indicate that AgNPs may affect the maturation and ovulation of preovulatory follicles and subsequently the process of egg laying.

**Abstract:**

The continuous development of poultry production related to the growing demand for eggs and chicken meat makes it necessary to use modern technologies. An answer to this demand may be the use of nanotechnology in poultry farming. One of the promising nanomaterials in this field are silver nanoparticles (AgNPs), which are used as disinfectants, reducing microbial pollution and the amounts of greenhouse gases released. This study aimed to evaluate the effect of AgNPs on the proliferation and apoptosis process in the granulosa cells of chicken preovulatory follicles. The in vitro culture experiment revealed that both 13 nm and 50 nm AgNPs inhibited the proliferation of the granulosa cells. However, a faster action was observed in 50 nm AgNPs than in 13 nm ones. A size-dependent effect of AgNP was also demonstrated for the caspase-3 activity. AgNPs 13 nm in size increased the caspase-3 activity in granulosa cells, while 50 nm AgNPs did not exert an effect, which may indicate the induction of distinct cell death pathways by AgNPs. In conclusion, our study reveals that AgNPs in vitro inhibit granulosa cell proliferation and stimulate their apoptosis. These results suggest that AgNPs may disrupt the final stage of preovulatory follicle maturation and ovulation.

## 1. Introduction

The challenge today is to provide the right amount of food for the ever-growing number of people in the world. This need can be met by sustainable agriculture, which aims to achieve high agricultural production, and its development depends on the implementation of new technologies, such as nanotechnology [1]. Nanotechnology increases the growth and agricultural production mainly due to the use of feed additives and due to its ability to provide proper hygiene in animal farms. Feed additives in the form of nano-minerals enable efficient absorption of nutrients for better utilisation of feed and other supplements, thus improving the quality and composition of animal products [2]. In the agricultural sector, poultry is one of the fastest-growing industries in the world due to the increasing demand for chicken eggs and meat [3]. Silver nanoparticles (AgNPs) are one of the most common nanomaterials and are used globally due to their high antimicrobial properties. AgNPs exhibit strong antibacterial activity against common poultry pathogens like *Escherichia coli* [4], *Salmonella* [5,6] and *Campylobacter jejuni* [6]. The elimination of pathogenic bacteria leads to better health of the poultry and improved quality of food obtained from them, resulting in increased productivity and farmer profit. Moreover, AgNPs are used in animal nutrition to reduce nitrogen oxides and ammonia emissions [7]. Concerning laying hens, the production efficiency is mainly related to the rate of laying. Egg production depends on the development of the ovarian follicles, which is a complex process regulated by gonadotrophins, neuropeptides, growth hormone, and other endocrine, autocrine, and paracrine factors [8]. Ovarian follicle development may also be regulated by external factors like environmental chemicals, including pesticides and industrial pollutants [9]. All these factors are involved in various ways in the regulation of granulosa and theca cell survival, apoptosis and proliferation [8]. In relation to the possibility of using AgNPs in poultry breeding, the question arises whether they could be among such factors and influence the process of proliferation and apoptosis in the chicken ovary. In our previous in vivo study, we showed that AgNPs affected steroid hormone concentrations in ovarian follicles as well as mRNA expression of genes encoding steroidogenic enzymes in the ovarian follicles in laying hens [10].

The studies conducted so far indicate that AgNPs exhibit significant toxicity to various types of human and animal cells. Decreased cell viability and anti-proliferative activity in response to AgNPs have been observed in neurons, keratinocytes, liver cells, lung epithelial cells, trachea epithelial cells and many others [11,12,13,14,15,16].

It is now believed that the toxicity of AgNPs is primarily related to the generation of reactive oxygen species (ROS) either on the surface of nanoparticles or by mitochondria and phagocytes [17,18]. High ROS levels are in turn associated with the induction of apoptosis, necrosis or autophagy [19]. Apoptosis is a tightly regulated process of physiological removal of superfluous, abnormal, infected, or damaged cells from the body. One of the most important components determining the course and execution of programmed death are enzymes called caspases [20]. The cytotoxic and apoptotic effects including the release of cytochrome c into the cytoplasm from the mitochondria, and the Bax translocation into the mitochondria were, for example, found in NIH3T3 fibroblasts under the influence of AgNPs [21]. Apart from the formation of ROS and mitochondrial dysfunction following AgNPs exposure, the depletion of the antioxidant glutathione in cells and superoxide dismutase, and the release of lactate dehydrogenase from cells, which occurs in the extracellular space in the event of cell membrane damage have been indicated [22,23,24,25]. Necrotic effects of AgNPs have been reported by many researchers in different cell types [25,26,27,28]. Necrosis is an uncontrolled cell death characterised by loss of membrane continuity, cytoplasm vacuolisation and spillage of intracellular contents into surroundings. Necrosis is related to the induction of the local inflammatory process and is independent of caspase activation [29]. Another type of cell death found recently under the influence of AgNPs is autophagic cell death preceded by autophagy dysfunction [30,31]. Autophagy is an evolutionarily conserved process that involves by lysosomal destruction of proteins or whole organelles [32].

In this study, we hypothesize that AgNPs affects proliferation and apoptosis in chicken granulosa cells in vitro. Hence, we evaluated the degree of proliferation and caspase-3 activity using nanoparticles of different sizes and concentrations. This approach allows us to investigate not only the possible influence of AgNPs on the function of ovarian follicles but also whether their size and concentration are important factors in this context.

## 2. Materials and Methods

The experiment was performed on 25-week-old “Hy-Line Brown” hens (n = 6) with an average body weight of 1.67 ± 0.16 kg. The birds were kept in a 14L:10D light regime, in individual cages, with free access to water and feed. The egg-laying time was recorded on each day for 3 weeks before the experiment to predict the time of ovulation. The experiments and animal procedures were approved by the First Local Animal Ethics Committee in Krakow, Poland (Approval no. 9/2015).

### 2.1. Cell Culture

Three of the largest preovulatory follicles (F3 < F2 < F1) were dissected from the ovary (n = 6) of chickens decapitated at the same stage of the ovulatory cycle (i.e., 22 h before ovulation) and placed in ice-cold phosphate-buffered saline (PBS, pH = 7.4) with the addition of an antibiotic-antimycotic solution (10,000 units penicillin, 10 mg streptomycin and 25 mg amphotericin B/mL). Granulosa layers separated from individual preovulatory follicles by Gilbert’s method [33] were transferred to separate beakers with sterile PBS solution following the stage of follicular development. Once washed, the fragments of the granulosa layer were cut into smaller fragments with scissors. After the digestion with 0.3% collagenase II at 37 °C for 10 min, the cell inoculum was filtered through gauze into a Falcon tube and centrifuged for 5 min at 250× *g*. The cell inoculum was suspended in M199 medium with 10% FBS and 0.2% antibiotic. Cells were counted in an automatic cell counter (Countess Automated Cell Counter, Invitrogen, Korea), and their viability (>85%) was determined using trypan dye blue. Then cells were seeded in a 96-well plate in an amount of 50,000 cells per well or 70,000 cells per well to determine the degree of proliferation and caspase-3 activity, respectively. Initially, the cells were cultured for 48 hours at 39 °C until attachment. Then the medium was changed to M199 with the addition of 5% FBS and 0.2% antibiotic, and the cells were cultured for another 30 hours with the addition of AgNPs ( Tusnovics Instruments SP., Krakow, Poland) of two different sizes: 13 nm and 50 nm. The following AgNP concentrations were used: 0.1 µg/mL, 1 µg/mL, 5 µg/mL or 10 µg/mL. These concentrations are in the range of minimum inhibitory concentrations (MIC) of AgNPs against pathogenic bacteria according to Arora et al. [34]. The cultures were maintained at 39 °C in a humidified atmosphere containing 5% CO_2_.

### 2.2. Proliferation Assay

The proliferation ratio of granulosa cells was assessed by the WST test (Quick Cell Proliferation Colorimetric Assay Kit Plus, BioVision, Milpitas, CA, USA) following the manufacturer’s protocol. After a 24-hour incubation of the granulosa cells with AgNPs, 10 µL of WST reagent was added to the medium and incubated under standard culture conditions for 4 h. After this time, absorbance was measured on a spectrophotometer (BioTek Epoch2, BioTek Instruments Inc., Winooski, VT, USA) at 440 nm wavelength.

### 2.3. Caspase-3 Activity Assay

Caspase-3 activity was determined by fluorimetric test (Caspase-3/CPP32 Fluorometric Assay Kit, BioVision, Milpitas, CA, USA) according to the manufacturer’s protocol. After 24-hour incubation of granulosa cells with AgNPs, the medium was collected, and cells were suspended in 50 µL of lysis buffer and left on ice for 10 min. Then 50 µL of reaction buffer (containing 10 mM DTT) and 5 μL of 1 mM DEVD-AFC substrate were added to each well. The plates were incubated for 2 h at 37 °C. Fluorescence measurement was performed on a fluorescence reader (BioTek FLx-800TBI, BioTek Instruments Inc., Winooski, VT, USA) using a 400 nm excitation filter and 505 nm emission filter.

### 2.4. Statistical Analysis

Statistical analysis of the results was carried out using the SAS statistical package (version 9.4). Before proceeding with the analysis, normal distribution was tested using the UNIVARIATE procedure. Due to the lack of normality of the proliferation data distribution, the non-parametric Kruskal–Wallis test was used. This test verifies the null hypothesis that all medians in the analysed groups are equal. If the null hypothesis was rejected, the procedure of multiple comparison of mean ranks with the Bonferroni adjustment was used. The results are presented in the form of a box-whisker chart showing the mean ranks and the first and third quartiles, as well as the minimum and maximum. The caspase-3 activity experiment was arranged as a two by three factorial design plus a control, so two statistical models were used. The first statistical model included all treatments and group effect as the classifying variable. In the case of significant effect of group (*p* < 0.05), the means were separated using the PDIFF option in SAS with the Tukey–Kramer adjustment. In the second model, the significance of the AgNPs size and concentration was tested. The statistical model included the Ag effect, concentrations and interaction between these effects as classifying variables, and it did not include a control group. AgNPs concentration effects were interpreted based on planned polynomial comparisons (polynomial contrasts) and associated linear and nonlinear changes of investigated parameters. In the case of a significant interaction (*p* ≤ 0.05) between the size and the dose of the AgNPs, the dose effect was analysed separately for each size of the nanoparticles. The results are presented as means and standard errors (SE). Figures were created using Grapher 11.0 (Golden Software Inc., Golden, CO, USA).

## 3. Results

### 3.1. Cell Proliferation

The results of the WST test showed a dose-dependent effect of AgNPs on the proliferation of granulosa cells obtained from F1, F2 and F3 preovulatory hen follicles. The smallest AgNP concentrations (0.1 and 1 μg/mL) did not affect cell proliferation, while higher concentrations (5 and 10 μg/mL) significantly reduced it (*p* < 0.0001) (Figure 1a–c). Granulosa cells in the control group and those treated with 0.1 μg/mL AgNPs for 24 h were polyhedral and showed continuous intercellular contact. Treatment of the cells with AgNPs at a concentration of 10 μg/mL for 24 h led to cell rounding and loss of communication between them. Cell culture for 30 h in the presence of 1 μg/mL AgNPs also had an adverse effect on the granulosa cells that manifested as rounding of the cells (Figure 2). This effect was especially noticeable with 50 nm AgNPs as opposed to 13 nm particles, where vestigial communication between cells was observed (see Appendix A).

### 3.2. Caspase-3 Activity

The statistically significant effect of nanoparticle size (*p* < 0.001) and concentration (*p* < 0.01) on caspase-3 activity was found in granulosa cells of preovulatory follicles (F3–F1) (Table 1). The 13 nm AgNPs at a concentration of 1 µg/mL increased the caspase-3 activity in granulosa cells of F3, F2, and F1 1.8, 2, and 2.2 times, respectively (*p* < 0.001). A similar effect was observed when the granulosa cells were exposed to 13 nm AgNPs at a concentration of 5 µg/mL. This dose increased caspase-3 activity 1.7 (F3), 1.4 (F2) and 2.7 times (F1) (*p* < 0.05). In turn, 50 nm AgNPs did not significantly influence caspase-3 activity.

## 4. Discussion

The production success of the laying hens depends on the efficient functioning of the bird’s reproductive system, specifically on the processes related to the production and development of ovarian follicles, i.e., the processes of proliferation and apoptosis. Numerous studies indicate that silver nanoparticles influence these processes in various cells. In the context of the use of AgNPs in poultry production, the question arises whether they can affect the proliferation and apoptosis of the granulosa cells of ovarian follicles, and, consequently, the number of eggs laid. The present study showed, for the first time, that regardless of the stage of chicken preovulatory follicle development, the effect of AgNPs on the granulosa cell proliferation in vitro, measured 24 h after exposure to these particles, did not differ significantly between follicles. However, it was clearly dose dependent. Microscope images taken after 24 h of treatment with AgNPs showed that the two highest doses of AgNPs were cytotoxic, and after 30 h of cell culture, the dose of 1 μg/mL showed an unfavourable effect. Concentration and time-dependent reductions in cell viability have been observed in multiple in vitro experiments, indicating that AgNPs potentially affect cell survival [12,17,34,35]. Furthermore, it was found that the smaller AgNPs are more toxic than the larger ones [22,36]. However, our microscopic observations after 30 h of incubation with 1 μg/mL of nanosilver indicated that 50 nm AgNPs were able to kill cells faster than smaller nanoparticles. Similar results were obtained by Yuan et al. [37] studying the impact of hydroxyapatite nanoparticles (HAPN) on the viability of the liver cells of the HepG2 line. The 45 nm HAPN nanoparticles showed a greater anti-proliferative effect than 26 nm HAPN. The authors explained these results by distinct endocytosis of different-sized nanoparticles, which is related to the cell membrane construction being an individual feature of each cell. The 50 nm nanoparticles were captured by the mammalian cells much faster and in greater amounts than those of other sizes [38,39]. Smaller nanoparticles tend to aggregate during cellular uptake, which limits their binding to the cell membrane.

Herein, the size-dependent effect of AgNPs was also observed for caspase-3 activity. Increased activity of this enzyme was found in the granulosa cells following their exposure to 13 nm AgNPs at the dose of 1 and 5 μg/mL, while 50 nm nanoparticles did not affect this activity. An increase in caspase-3 activity in tumour cell lines under the influence of AgNPs was also observed by Arora et al. [34] and Bin-Jumah et al. [35]. Numerous studies have been suggested that the main mechanism of AgNPs toxicity is related to oxidative stress associated with the increased production of ROS in cells [17,30,31,34,36]. Oxidative damage involves cellular macromolecules, including nucleic acids, phospholipids and proteins. Mitochondrial DNA is especially vulnerable to ROS attack due to its proximity to the respiratory chain, which is the main site of peroxide production, and the lack of protective histones [40]. ROS lead to cell death by apoptosis through the activation of caspases. It has been suggested that it is mainly smaller silver nanoparticles (15–20 nm) that induce ROS formation [20,41], which may be the reason for an increase in caspase-3 activity in the granulosa cells after incubation with 13 nm AgNPs. In the experiment conducted by Tang et al. [42], ROS formation was also greater in rat tracheal epithelial cells treated with 10 nm AgNPs as compared to 100 nm ones. Similarly, Kim et al. [25] found that the smallest AgNPs (10 nm) had a greater potential to induce apoptosis in the MC3T3-E1 cells than larger ones (50 nm, 100 nm).

The studies on nanoparticle toxicity indicate that they can be lethal to cells by activating other pathways not related to the apoptotic process. The necrotic effects of AgNPs have been reported in many studies [13,25,34]. Whether AgNPs induce apoptosis or necrosis depends on many factors, e.g., exposure time, the concentration of nanoparticles, cell type [16]. Another example of cell death is a mitotic catastrophe that results from abnormalities taking place during mitosis, namely the lack or delayed entry of the cell into the G1/S phase or cell cycle arrest in the G2 phase [43]. Intestinal cells that were stopped in the G2 phase under the influence of zinc oxide nanoparticles were observed by Setyawati et al. [44]. Asharani et al. [12] reported that silver nanoparticles induced cell arrest in the G2 phase and damaged DNA in human glioblastoma cells and fibroblasts. Anti-proliferative action of AgNPs associated with a cell cycle arrest in the S phase in the mouse macrophage cell line was shown by Park et al. [45], and it was observed in the lung adenocarcinoma cell line, regardless of ROS generation [46].

Cells can also die because of autophagy. The process induced by AgNPs was observed, among others, in human hematopoietic cells [47], the cervical cancer cell line [48], the human pancreatic ductal adenocarcinoma [49], and in the mouse hippocampal neuronal cell line [31]. Summarising the above considerations, the type of cell death may depend on the size of the silver nanoparticles; smaller AgNPs (13 nm) may induce apoptosis, while larger ones may cause necrosis, mitotic catastrophe, or autophagic cell death. However, further experiments are needed to verify this hypothesis.

In conclusion, AgNPs inhibit the proliferation of the granulosa cells isolated from the chicken preovulatory follicles, and the effect of their action depends on the nanoparticle size. The 50 nm AgNPs seem to inhibit proliferation faster than 13 nm ones. The 13 nm AgNPs increased caspase-3 activity in granulosa cells, while 50 nm AgNPs do not affect the activity of this enzyme, which may indicate that they induce distinct cell death pathways. The results obtained suggest that AgNPs may disrupt the final stage of growth and development of preovulatory follicles by inhibiting granulosa cell proliferation and causing their apoptosis. It also cannot be excluded that AgNPs may affect the ovulatory process of the largest follicle.

## Figures and Tables

**Figure 1 animals-11-01652-f001:**
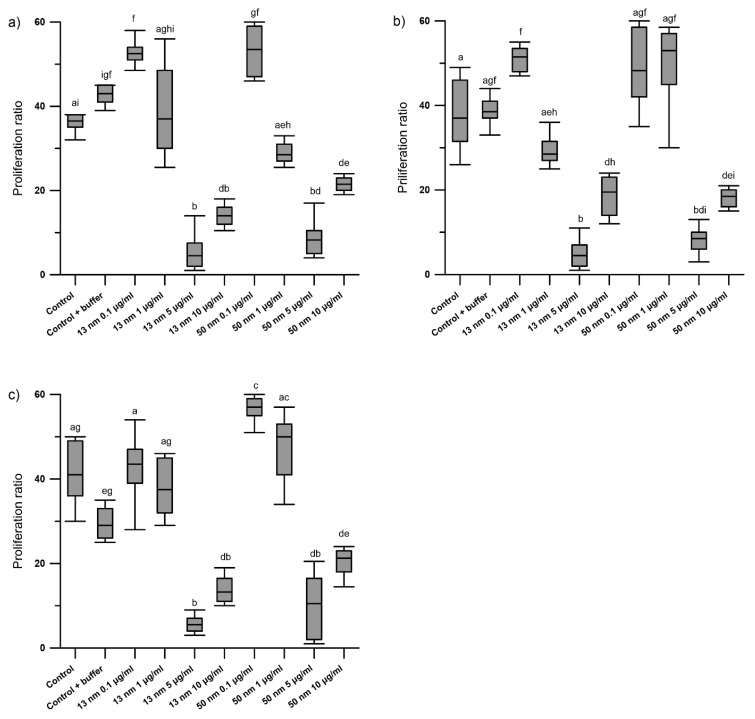
In vitro effect of increased concentrations of 13 nm and 50 nm AgNPs on the proliferation ratio of chicken granulosa cells isolated from F3 (**a**), F2 (**b**), and F1 (**c**) preovulatory follicles after 24 h of incubation. Values marked with different letters differ significantly (*p* < 0.05).

**Figure 2 animals-11-01652-f002:**
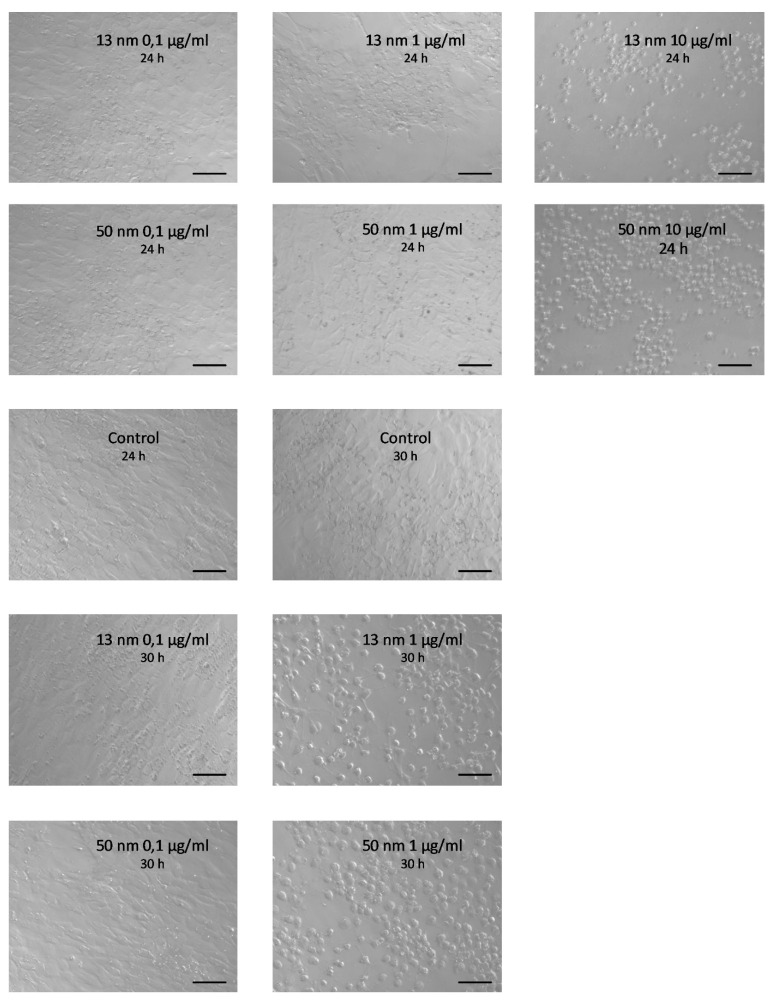
Representative phase-contrast micrographs (taken with an Axio Observer from Zeiss, Germany) showing chicken granulosa cell culture derived from F2 preovulatory follicles with the addition of increased concentrations of 13 nm and 50 nm AgNPs taken after 24 h and 30 h of the incubation period. For better visualization of the difference between 1 µg/mL 13 nm AgNPs and 1 µg/mL 50 nm AgNPs after 30 h of incubation, see Appendix A. Scale bars = 50 µm.

**Table 1 animals-11-01652-t001:** Caspase-3 activity in the granulosa cells isolated from preovulatory (F3-F1) follicles after 24 h of incubation with increased concentrations of 13 nm and 50 nm AgNPs.

Parameter	Control	Control + Buffer	Ag	Dose			SE	Group	Control vs. Control + Buffer	Ag	Dose	Ag × Dose	Dose (Trend)	
				0.1 µg/mL	1 µg/mL	5 µg/mL							Linear	Quadratic
Caspase-3														
F1	36.20 ^c^	35.20 ^c^	13 nm	44.20 ^c^	76.80 ^ab^	96.20 ^a^	3.623	<0.001	0.756	<0.001	0.001	0.001	<0.001	0.456
			50 nm	49.60 ^c^	44.20 ^c^	52.00 ^bc^							<0.005	0.219
F2	28.80 ^c^	37.20 ^bc^	13 nm	43.40 ^bc^	73.60 ^a^	53.80 ^ab^	2.513	<0.001	0.113	0.001	0.019	0.095	0.123	0.015
			50 nm	45.00 ^abc^	46.40 ^abc^	35.40 ^bc^								
F3	25.60 ^de^	33.20 ^cd^	13 nm	41.20 ^abc^	61.20 ^a^	56.00 ^ab^	2.456	<0.001	0.083	<0.001	0.003	<0.001	0.135	0.004
			50 nm	38.60 ^bcd^	36.00 ^bcd^	19.60 ^e^							<0.001	0.638

^abcde^ means with different letters differ significantly (*p* ≤ 0.05).

## Data Availability

All data, methods, and results of statistical analyses are reported in this paper. We welcome any specific inquiries.

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
