# Peer review of "Effects of Silver Nanoparticles on Proliferation and Apoptosis in Granulosa Cells of Chicken Preovulatory Follicles: An In Vitro Study"

_animals, 2021, doi:10.3390/ani11061652_

Round 1

Reviewer 1 Report

The study of Katarzyńska-Banasik and colleagues evaluated effects of 13nm and 50nm silver nanoparticles (AgNPs) on proliferation and caspase-3 activity in granulosa cells of chicken preovulatory follicles. By using a WST test kit, a Caspase-3 Assay and microscopic analysis the authors studied cell viability, caspase-3 activity and cell morphology of cultured chicken granulosa cells from three of the largest preovulatory follicles (F3<F2<F1) after 24 and 30 hrs treatment with different AgNP concentrations.

The authors found that 13 nm and 50 nm AgNPs inhibited proliferation and changed cell morphology in a dose dependent manner, but that only 13 nm particles increased the caspase-3 activity. From their data they concluded that AgNPs may disrupt the final stage of preovulatory follicle maturation and ovulation.

In light of growing application of AgNPs as antibacterial agents in poultry production the study is interesting and relevant. The experimental design is appropriate, the experiments seem properly done and the data statistically evaluated in an appropriate way.

However, the before publication the manuscript has to be substantially revised because of inconsistencies and incomprehensible presentation and interpretation of the data.

Abstract:

1)

Contradictory statements: “…this effect was faster for 13 nm nanoparticles…” (SIMPLE SUMMARY)

“   a faster action of 50 nm AgNPs than 13 nm ones was observed” (ABSTRACT)

Results

2)

Table 2 is not readable. It must be re-formatted.

3)

Insert bars in Figure 2 instead of only indicating 40X magnification in the legend. In addition, it should be indicated, which microscope and optics (Nomarski, phase contrast etc.) have been used.

4)

It would be helpful to the reader to know which AgNP concentrations are used in poultry farms and, if known, which concentration are actually found in the animals and in particular in ovarian tissue.

5)

Line 128: wrong citation, should be 32!

6)

Lines 204-206: The statement “This effect was especially noticeable with 50 nm AgNPs as opposed to 13 nm particles…” is not comprehensible from Figure 2. Higher magnification or better quality might help.

Discussion

7)

Lines 18-19: The statement “However, our microscopic observations indicated that 50 nm AgNPs were able to kill cells faster than smaller nanoparticles” is not comprehensible on the basis of the data shown.

8)

The discussion on different mechanisms of cell death (apoptosis vs. necrosis) is pointless as long as cell death is not shown and not comprehensible on the basis of the data presented (e.g. lines 63,64).

9)

Line 68: Which data show that “…50 nm AgNPs inhibited proliferation faster than 13 nm ones”? This is neither shown in Figures 1 nor 2.

Reviewer 2 Report

The authors, essentially support their conclusions on two results: the caspase 3 response and the cell proliferation assay. However, they also point out that other cell death pathways can lead to the same results when silver nanoparticles with such size and concentration characteristics are present. Therefore, I suggest that the authors should perform other tests to support their conclusions.

Reviewer 3 Report

The topic of the presented paper could be of interest.

However, several concerns need to be solved.

  • Ethical statement is lacking
  • the relevance of the examined concentrations to a real exposure has not been sufficiently clarified
  • other parameters have to be examined to drawn the reported conclusions. In particular, the authors examined the effect on cell viability. Cell proliferation has to be examined in other way

Round 2

Reviewer 2 Report

I have no more recommendations to make.

Reviewer 3 Report

My previous criticisms have not been completely addressed.

In particular, the rationale for the examined concentrations has not been sufficiently detailed.

Moreover, the study does not demonstrate effect on cell proliferation